# Identification of the Efficacy of Ex Situ Conservation of *Ammopiptanthus nanus* Based on Its ETS-SSR Markers

**DOI:** 10.3390/plants12142670

**Published:** 2023-07-17

**Authors:** Jingdian Liu, Xiyong Wang, Ting Lu, Jiancheng Wang, Wei Shi

**Affiliations:** 1State Key Laboratory of Desert and Oasis Ecology, Key Laboratory of Ecological Safety and Sustainable, Development in Arid Lands, Xinjiang Institute of Ecology and Geography, Chinese Academy of Sciences, Xinjiang 830011, China; ariiiiiink@gmail.com (J.L.); wangxy@ms.xjb.ac.cn (X.W.); 2College of Forestry and Landscape Architecture, Xinjiang Agricultural University, Xinjiang 830011, China; luting0909@126.com; 3Turpan Eremophytes Botanic Garden, The Chinese Academy of Sciences, Xinjiang 838008, China

**Keywords:** *Ammopiptanthus nanus*, ex situ conservation, ETS-SSR

## Abstract

*Ammopiptanthus* possesses ancestral traits and, as a tertiary relict, is one of the surviving remnants of the ancient Mediterranean retreat and climate drought. It is also the only genus of super xerophytic, evergreen, broad-leaved shrubs. *Ammopiptanthus nanus*, one of the two species in this genus, is predominantly found in extremely arid and frigid environments, and is increasingly threatened with extinction. Study of the species’ genetic diversity is thus beneficial for its survival and the efficacy of ex situ conservation efforts. Based on transcriptome data, 15 pairs of effective EST-SSR were screened to evaluate *A. nanus* genetic diversity. In all, 87 samples from three populations were evaluated, the results of which show that ex situ conservation in the Wuqia region needs to be supplemented. Conservation and breeding of individual *A. nanus* offspring should be strengthened in the future to ensure their progeny continue to exhibit high genetic diversity and variation.

## 1. Introduction

*Ammopiptanthus*, an oligotypic legume and Tertiary relict plant of the ancient Mediterranean, is the only super xerophytic, evergreen, broad-leaved shrub endemic to desert areas in central Asia [1]. It includes two species, namely, *Ammopiptanthus nanus* (M. Pop.) Cheng f. and *Ammopiptanthus mongolicus* (Maxim. ex Kom.) Cheng f., both of which are listed in the *China Plant Red Data Book* as a Rare and Endangered Plant, and also listed as critically endangered in the IUCN Red List Category and Criteria [2,3]. The enormous root structure of *A. nanus* is physiologically and ecologically adapted to difficult environments, allowing it to live on barren mountains and the stony Gobi under significant drought conditions by preventing wind and fixing sand [4]. Temperatures in the distribution area of *A. nanus* frequently reach over 35 °C in the summer and −20–30 °C or lower in the winter. The process of soil humus accumulation is weak in the region, and the soil has low fertility; therefore, the sand holly has acquired traits such as early resistance, cold resistance, infertility resistance, and a developed root system [5]. According to the experience of herdsmen, running the branches of *A. nanus* through a camel’s nose can prevent infection [6]. Additionally, the shrub’s broad-leaved, evergreen character in desert areas as well as its thick leaves, golden yellow corolla, and short plant body lend it utility as an ornamental flower in the spring [7]. As the raw material for extracting industrial oil containing various special alkaloids and flavonoids in its stems, branches, leaves, and seeds, *A. nanus* has great value for research and development in the biological industries [8]. Various quinolizidines including lupanine, sparteine, and their derivatives have been isolated from the leaves of *A. nanus* [9]. *A. nanus* is a relatively primitive legume species belonging to the third generation of surviving remnant plants along the coast of the ancient Mediterranean Sea prior to climate-driven drought and retreat [10]. Thus, *A. nanus* also provides an important scientific basis for studying climatic characteristics, geographic effects of environmental change, plant migration, and even the origin and formation of desert vegetation in central Asia during the Tertiary period [11]. *A. nanus* grows in a dryer and cooler habitat than that of *A. mongolicus*. Therefore, the former has a shorter habit, usually with one-foliolate leaves, conspicuous leaf venation, a thicker root cortex, a more complex karyotype, and more vulnerable phyto-communities [12,13]. *A. nanus* is primarily found in arid and frigid environments with fragile ecologies and generally more challenging environmental conditions [14]. *A. nanus* has not undergone successful ex situ preservation as an outcome of climate and geographical changes and other ecological characteristics, and its distribution area is quickly diminishing.

For these reasons, it is urgent to take effective measures to protect *A. nanus*. There are two methods of plant conservation: in situ and ex situ. The latter is a more reasonable method for the wide distribution and objective environment of *A. nanus*, and may also serve as an effective research condition for move conservation, a practice which has become an important component of global biodiversity conservation initiatives [15,16]. The practice is intended for the protection of wildlife species in areas outside their native community while maintaining the normal growth and reproduction of the species in the protected area [17]. Ex situ conservation is a supplement to in situ conservation. When the original habitat of a species is destroyed due to natural or man-made causes, or the population of the species is very low, ex situ conservation serves as a more effective means to protect the species [18]. Through ex situ conservation, scientists and practitioners can better understand the morphological characteristics, systematic and evolutionary relationships, and growth and development of conservation organisms, thus providing a stronger basis for in situ conservation and monitoring of wild species and laying a foundation for their population reconstruction [19]. Several interventions involving ex situ management were anticipated to provide a >50% likelihood that the species in question would persist; however, these actions were more expensive than in situ alternatives alone. It was predicted that the anticipated benefits of ex situ measures would be negated by the greater uncertainty and randomness associated with establishing and maintaining ex situ populations [20]. Ex situ conservation that is employed naively can produce ineffective management outcomes. Thus, successful ex situ protection requires rigorous planning that precisely analyzes the objectives, management alternatives, and the likelihood of success of the conservation program.

Genetic diversity is the foundation of biological diversity and the basis for maintaining a species’ reproductive vitality, resistance to pests and diseases, and ability to adapt to environmental changes [21]. It is also important for breeding programs seeking to improve desirable traits and create new varieties. Genetic diversity lays a foundation for evolution and adaptation to the environment. The richer a population’s genetic diversity, the stronger its adaptability to environmental changes [22,23]. Genetic diversity is not only closely related to the long-term vitality and reproductive capacity of biological species [24], but is also important for the introduction and domestication of plants and gene preservation [25]. Previous research has found that the traits of *A. nanus*, like those of other plants, vary across different regions due to environmental factors [13]. Understanding such environmental effects of genotype–environment (GxE) interactions and associated problems and opportunities is critical for efficient breeding program design and genetic material deployment. The various consequences of (GxE) interactions determine trait phenotypes at the level of plant biological organization. The impacts of rainfall and temperature on blackcurrant fruit quality, for example, were revealed by merging metabolomics and sensory investigations and established the groundwork for selecting high-quality blackcurrant cultivars and the constructing suitable production sites [26]. In addition, GxE can be used to screen for appropriate environmental conditions in order to maximize the genetic benefits of forest tree breeding [27]. This shows there is still a lot of space for growth in this field.

SSR consists of short (generally 1–6 bp) tandem replication of nucleotide motifs and is characterized by a high degree of variation, codominant inheritance, and wide distribution. SSR is recognized as one of the best molecular markers [28], and its simple operation is a common means of analyzing molecular genetic diversity [29]. SSR markers are superior to SNP and AFLP markers for detecting polymorphism because SSR is an effective multiple-allele marker, whereas SNP and AFLP are single-allele markers [30]. The features and co-dominance of SSR markers make them the preferred DNA markers for assessing plant genetic diversity [31]. In particular, the highly conserved nature of EST-SSR makes it more accurate than G-SSR at predicting genetic differences and more generalizable across similar species [32,33].

In this study, we utilized EST-SSR markers developed from the transcriptome genes of *A. nanus* to assess the efficacy of ex situ conservation efforts. By investigating the differences in genetic diversity between the original population in Wuqia and an ex situ population at the molecular level, we were able to identify the impact of ex situ conservation on *A. nanus*. Thus, the study provides a theoretical foundation for the ex situ conservation, breeding, and inheritance of *A. nanus*.

## 2. Results

### 2.1. Library Quality Management, Data Sequencing and Quality Assurance

Following the preceding series of analyses and quality checks, we obtained 14.57 GB of clean data, and the proportion of Q30 bases in each sample did not fall below 92.77%. Two samples of clean data are presented in Table 1.

### 2.2. Transcriptome Sequencing Library Data Assembling

Qualified transcriptome sequencing libraries are essential for reliable transcriptome data analysis results [34]; the clean data from the two samples were compared with the assembled Transcript or Unigene libraries [35]. The statistical results of this comparison are presented in Table 2.

### 2.3. Transcriptome Sequencing Library Quality Assessment

To assure the quality of transcriptome sequencing libraries, these were evaluated from three distinct vantage points. The randomness of mRNA fragmentation and mRNA degradation were assessed by examining the distribution of insert fragments on Unigene. After mRNA fragmentation, the insert fragment size selection can be interpreted as the random extraction of subsequences from mRNA sequences. If the sample size (number of mRNAs) is larger, the interruption method and time control are more suitable, and the probability of each part of the target RNA being extracted is closer. That is, the randomness of mRNA fragmentation is greater, and the reads coverage on mRNA is more uniform. The degree of randomness of mRNA fragmentation was examined by simulating the results of mRNA fragmentation through the distribution of mapped reads in Unigene. Each sample’s positional distribution of mapped reads on mRNA transcripts was obtained (Figure 1).

The horizontal axis in Figure 1 represents the mRNA position, while the vertical axis represents the percentage of reads in mapped reads within the corresponding position interval. Because the lengths of the reference mRNA varied, each mRNA was divided into 100 intervals based on its length, and the number and proportion of mapped reads in each interval was determined. During library preparation, the dispersion of insert lengths reflects the effect of gel cutting or magnetic bead purification (Figure 2). The horizontal axis represents the distance between the starting and ending points of the double-ended reads in the Unigene library, ranging from 0 to 800 bp, and the vertical axis represents the number of double-ended reads or insert fragments at various distances between the starting and ending points of the comparison.

Finally, the library capacity and the sufficiency of reads (mapped reads) relative to the Unigene library was assessed by plotting the saturation. Sufficient valid data are required for accurate analysis. There is a positive correlation between the number of genes detected by transcriptome sequencing and the amount of sequencing data; the greater the quantity of sequencing data, the greater the number of genes detected. However, the number of genes in a species is limited, and gene transcription is time- and space-specific, so the number of detected genes tends to saturate as the sequencing quantity increases. To determine if the data are sufficient, it is necessary to determine if the number of newly detected genes decreases with increasing quantity of sequencing data and signaling saturation. Using mapped reads for each sample, the saturation of the number of detected genes was simulated and depicted on a graph (Figure 3).

Saturation curves were generated by dividing mapped reads into 100 equal portions and progressively increasing the number of genes detected by viewing the data. The horizontal axis represents the number of reads (in 10^6^ units), while the vertical axis represents the number of genes detected (in 10^3^ units). If the slope gradually flattens as the curve extends to the right, it indicates that as the amount of sequencing data increases and tends to saturate with an adequate amount of valid sequencing data, each sample contains fewer and fewer newly detected genes.

### 2.4. Annotation of Unigene Function

Unigene Orthology results were obtained using the BLAST [36] software to compare Unigene sequences with the NR [37], Swiss-Prot [38], GO [39], COG [40], KOG [41], eggNOG4.5 [42], and KEGG [43] databases, and by obtaining KEGG in KEGG using KOBAS2.0 [44]. After predicting the amino acid sequence of Unigene, the software HMMER [45] was utilized to compare it with the Pfam [46] database in order to obtain Unigene’s annotation information.

NCBI’s NR database is a non-redundant protein database that contains the Swiss-Prot, PIR (Protein Information Resource), PRF (Protein Research Foundation), and PDB (Protein Data Bank) protein databases, as well as translated CDS data from GenBank and RefSeq. The EBI (European Bioinformatics Institute) maintains the Swiss-Prot database, which contains a highly reliable database of annotated protein information with relevant references and revision. The GO (Gene Ontology) database is a standardized international gene function classification system that offers a dynamically updated standard vocabulary for describing the functional properties of genes and gene products in an organism. The Molecular Function, Cellular Component, and Biological Process categories of the Gene Ontology (GO) database each describe the molecular functions that a gene product may execute, as well as the cellular environment and biological processes in which it is involved. The COG (Clusters of Orthologous Groups) database is a homology classification database for gene products and an early database for identifying direct homologous genes, based on extensive comparisons of protein sequences from numerous organisms. The KOG (euKaryotic Orthologous Groups) database is intended for eukaryotes and classifies homologous genes from various species into different orthologous clusters based on direct homology of genes combined with evolutionary relationships; KOG has 4852 classifications at present. Genes from the same orthologous clusters have the same function, allowing functional annotations to be directly inherited by other KOG cluster members. The eggNOG (v4.5) database contains functional descriptions and functional classifications of immediate homologous proteins, combining COG, KOG, and many more proteins; it covers significantly more protein sequences than COG and KOG, and it includes 5228 viral proteins. The KEGG (Kyoto Encyclopedia of Genes and Genomes) database is a database that comprehensively analyzes the metabolic pathways and functions of gene products in cells. It includes metabolic pathways (PATHWAY), pharmaceuticals (DRUG), diseases (DISEASE), gene sequences (GENES), and genomes (GENOME). Use of this database facilitates the investigation of gene and expression data as a holistic network.

The Pfam (Protein family) database is the most comprehensive classification system for annotating protein structural domains by constructing HMM statistical models from the amino acid sequences of each family via protein sequence alignment. Proteins are composed of one or more structural domains, which are functional regions, and exhibit sequence conservation. The functions of proteins can be predicted based on the sequence of structural domains. Selecting a BLAST parameter E-value that was ≤1 × 10^−5^ and an HMMER parameter E-value that was ≤10^−10^ yielded a total of 39,437 Unigene sequences with annotation information for this research (Table 3).

### 2.5. EST-SSR the Select of Primers

*A. nanus* gene sequences were screened using MISA (Table 4). In a transcriptome genome of approximately 42.52 Mb, there were 11,645 SSR marker repeats with a total length of 169,920 bp. The frequency of SSR markers in the gene sequence was 273.88 SSRs/Mb, with a density of 3996.48 bp/Mb, and the transcriptome genome’s total SSR content was 0.4%.

Figure 4 summarizes the results, classified by length of the SSR loci. All loci ranged in length from 10 to 42 bp, with an average of 14.59 bp. Overall, 10 bp was the most common repeating unit, occurring 1945 times. It was followed by 15 bp, 12 bp, and 18 bp, with respective frequencies of 1634, 1447, and 1391.

The statistical analysis of the number of repeats of SSR loci (Figure 5) showed that all repeats fell within the range of 5 to 24. The maximum proportion of duplicates among them was 10, accounting for 18.78% of the total. This was followed by 5, 6, and 11 times, representing 13.39%, 12.52%, and 10.57% of the duplicates, respectively.

The repetitive unit classes of SSR loci are summarized in Table 5. The total length of the SSR sequence was 169,920 bp, with an average length of 14.59 bp. The most prevalent nucleotide type is the single-nucleotide repeat (57.8%), with an average length of 13.17 bp and a total length of 88,629 bp. The remaining sequences are trinucleotide repeats (20.4%), dinucleotide repeats (19.16%), tetranucleotide repeats (2.15%), and pentanucleotide repeats (0.28%), with lengths of 39,996 bp, 34,494 bp, 5088 bp, and 858 bp, respectively, and average lengths of 16.84 bp, 15.46 bp, 20.35 bp, 25.91 bp, and 34.32 bp, respectively. The SSR sequences contained a total of 238 nucleotide repeats; Table 6 lists the principal SSR sequence repeat units and the corresponding numbers of repeats.

The loci results for *A. nanus* are depicted in Figure 6, where the vertical axis represents the intensity of fluorescence, and the horizontal axis represents the size of SSR product fragments.

### 2.6. Genetic Diversity Analysis of A. nanus

Table 7 shows that the total genetic diversity (Ht) of the 15 primer pairs detected in 87 samples ranged from 0.691 to 0.904, with a mean value of 0.83. The mean observed heterozygosity (Mean Ho) ranged from 0 to 0.55, with a mean value of 0.25. The mean expected heterozygosity (Mean He) ranged from 0.218 to 0.716, with a mean value of 0.47. The ranges of the intra-population inbreeding coefficient (Fis),total population inbreeding coefficient (Fit), and genetic differentiation coefficient (Fst) are −0.021 to 1, 0.326 to 1, and 0.208 to 0.733, respectively. Gene flow (Nm) ranged from 0.091 to 0.951, with a mean value of 0.43. Values of Nm which are <1 indicate that differentiation between populations may occur due to genetic drift, and that there is a low level of gene flow among most populations. As shown in Table 7, F6 is the most efficient primer, followed by F3 and F1. F2 produces fewer effective results.

At the populations’ level, 15 pairs of SSR primers revealed the population genetic diversity of a total of 87 samples from the three populations (Table 8). The number of alleles (Na) ranged from 3.6 to 6.333, containing a total of 14.93, with a mean value of 4.978; the effective number of alleles (Ne) ranged from 2.429 to 2.714, with a mean value of 2.58; the Shannon’s information index (I) ranged from 0.827 to 1.121, with a mean value of 0.969; the observed heterozygosity (Ho) ranged from 0.231 to 0.29, with a mean value of 0.254; expected heterozygosity (He) ranged from 0.409 to 0.535 with a mean value of 0.466; genetic diversity parameters of the populations (PPB) ranged from 0.7333 to 0.9333, with a mean value of 82.22% and the highest, at 93.33%, in the Wuqia region. This indicates that the top priority for conservation of *A. nanus* genetic diversity is in the Wuqia region.

The genetic distance and genetic differentiation coefficient (FST) are important parameters for evaluating the degree of genetic difference and similarity among *A. nanus* populations. These were calculated among different populations of *A. nanus* based on Nei’s genetic distances using the GenAlEx software (Table 9). The genetic distances between populations of *A. nanus* ranged from 2.659 to 4.9, with the largest genetic distances between the Wuqia and Kyrgyzstan populations and the smallest between the ex situ and Kyrgyzstan populations. The genetic differentiation coefficients between populations ranged from 0.352 to 0.431, with the largest between the Wuqia and Kyrgyzstan populations and the smallest between the Wuqia and Kyrgyzstan populations. The genetic differentiation coefficient between populations ranged from 0.352 to 0.431, with the Wuqia and Kyrgyzstan populations having the largest genetic differentiation coefficient and the Wuqia and ex situ populations having the smallest genetic distance.

Molecular ANOVA of the genetic variation within and between populations of *A. nanus*, as shown in Table 10, indicates that 29% of the genetic variation in the 87 *A. nanus* germplasms collected from the three populations originated between populations, while nearly 71% of the genetic variation originated within populations (Figure 7). The population genetic structure was analyzed using Structure Harvester, and the mean value corresponding to each K value was calculated (Figure 8). The maximal inflection point occurred when K = 3, indicating that the population genetic structure had a theoretical population size of 3.

Visual coordinates and primary coordinate analysis were utilized to depict data similarity or dissimilarity for study sample populations. The data from 87 samples were displayed and processed to create PCoA scatter plots (Figure 9).

The results show that the three populations of *A. nanus* still belong to the same species but are extremely differentiated due to different natural conditions. There is substantial genetic variation both within and between populations, and within-population variation is greater. The high degree of genetic divergence across populations may be the result of genetic drift or the combined influence of natural selection and genetic differentiation induced by gene flow between places.

## 3. Discussion

Ex situ conservation conserves a species germplasm through artificial breeding in order to increase population sizes and prevent extinction in the wild [47]. There has been extensive work conducted in this area through methods like artificial seeding and propagation in botanical gardens and natural environments. However, these efforts can be expensive, and their efficacy has not been confirmed [48]. Inbreeding can lead to decreased fitness and thus endanger the survival of small populations [49] through what is known as inbreeding depression [50]. Genetically, this phenomenon may occur because favorable alleles at one locus are typically dominant, allowing rare and recessive deleterious alleles at the same locus to persist in a population [51]. The increased purity caused by inbreeding increases the likelihood of deleterious alleles occurring as pure heterozygotes. When this occurs, the effects of deleterious alleles cannot be concealed by dominant favorable alleles, resulting in inbreeding depression [52]. It appears that dominance is the most prevalent cause of inbreeding depression [53]. This has the potential to accelerate the extinction of *A. nanus* [54].

Due to the present scarcity of *A. nanus* populations, it is crucial to avoid inbreeding. Purifying deleterious alleles is one effective method for avoiding inbreeding depression [55]. Inbreeding increases the purity of alleles and, consequently, the likelihood that the associated deleterious traits will manifest. This may result in the elimination of deleterious alleles from congenic populations [56]. However, in extremely tiny populations, the effect of removal is also quite limited [57]. There are few direct indications from natural populations that purification can eradicate inbreeding depression [58]. Studies have shown that deleterious alleles are eliminated over time [59]. In the past, it has been proposed to eliminate deleterious alleles from small populations through artificial inbreeding. Nevertheless, many biologists believe the associated hazards outweigh the potential benefits [60]. SSR markers are valuable instruments for analyzing genetic diversity and genetic structure due to their high information density, homology, and multiple alleles [61]. Our predecessors conducted a genomic survey and SSR analysis of *A. nanus* homologs of *A. mongolicus* [62]. Several molecular investigations of *A. nanus* have also been conducted, including chloroplast genome comparisons, phylogenetic analyses [63], and RAPD analyses of genetic diversity [64]. In addition, it is possible to derive numerous specific markers associated with specific resistance traits, which provides a clearer indication of the level of kinship and genetic diversity among sampled materials [65]. SSR not only makes it possible to identify and evaluate the efficacy of ex situ conservation of *A. nanus*, but it also provides a valuable tool for future population genetics research and conservation initiatives.

In the Wuqia region, there are only a few populations and individuals of *A. nanus*, but their descendants exhibit a high level of genetic diversity and progeny variation. From the results of this study, they can thus be divided into two conventional protected areas, with the Kyrgyz subspecies in its own protected area. This study indicates that *A. nanus* has a high level of individual genetic diversity, so both in situ and ex situ conservation should be implemented for this population along with abundant genetic variation and rare alleles. *A. nanus* must also be conserved in situ to assure genetic diversity, as is the case with species that are widely distributed but rapidly declining [66]. The results of this investigation indicate that the ex situ conservation of Wuqia germplasms requires strengthening. Significant investments in protected areas such as the Kyrgyz area are required to reduce the risk of extinction. However, *A. nanus* has a very limited range, as evidenced by its likelihood of genetic isolation, genetic drift, and absence of genetic mutations [67]. Additionally, it is common for plants to decline due to hard inbreeding, particularly for such narrowly distributed species, so the potential for genetic diversity loss is high [68]. Inbreeding may be another major threat.

SSR analysis for *A. nanus* alone, on the other hand, has clearly identified the distinct communities of *A. nanus* generated by the environment. GxE has long been regarded as a critical component in plant breeding [69]. With the advent of genomics technology, several good predictors for the genotypic dimension of GxE interactions have been presented [70,71]. The ex situ conservation did not occupy all the germplasm resources, especially lacking the genotypes in Kyz. However, we possess the fundamental theory and methodology to improve the conservational efficiency of future work using more clarified evidence on GxE of *A. nanus.* Due to habitat devastation in the wild, ex situ conservation is an extremely effective conservation measure for species that are critically endangered. Establishing botanical gardens is also among the most effective methods for ex situ conservation of endangered species such as *A. nanus* in TEBG [72]. Nonetheless, the genetic diversity of ex situ plants in botanical gardens has been largely ignored or inadequately evaluated [73].

Given the increasing threats to genetic diversity posed by inbreeding and environmental change, genetic characterization is necessary to prevent the decline of these new populations and maximize their evolutionary potential [74]. While population genetic data are rarely incorporated into species assessments and management plans, this does not mean that conservationists are unwilling to incorporate genetics into their practices [75]. This suggests that, while researchers are most likely to conduct conservation genetics research, they are often unable to link policy and conservation outcomes in practice [76,77]. With more frequent collaboration between researchers and governments, there may be a larger role for genetics in future conservation programs. In this research, 87 samples of *A. nanus* from three populations were analyzed using ETS-SSR molecular markers to evaluate the efficacy of ex situ conservation and support further conservation and artificial breeding of this germplasm resource.

*A. nanus* remains on the verge of extinction. For comprehensive conservation of this germplasm resource, it is necessary to focus on the genetic diversity of ex situ plant offspring as well as the quantitative conservation of individuals. Doing so will ensure the reproductive renewal of this extremely small population. Additionally, it is important to strengthen government collaboration with botanical gardens by incorporating the results of research into resource-management policy. Effective cooperation will have a positive influence on the conservation and utilization of the unique genes of *A. nanus*. Future research and investment need to be increased to demonstrate the value of *A. nanus* for adapting to extreme environments.

Given the distributional characteristics of *A. nanus*, we make the following recommendations for its conservation in the future:The government should intervene to increase investment in in situ conservation and broadcast relevant policies. Seeds of different populations in *A. nanus* with high genetic diversity should be collected for propagation in order to obtain the maximum available genetic diversity.Plants from different populations should be introduced for appropriate crosses to increase the level of biodiversity of *A. nanus*. Further research on the reproductive biology of *A. nanus* is required to ensure the effective conservation of the species.

## 4. Materials and Methods

### 4.1. RNA Sample Detection and RNA Library Development

At the beginning of the study, one sample of *A. nanus* was collected from the Turpan Eremophytes Botanic Garden (CAS) (TEBG, 42.3° N 89.2° E) for transcriptome sequencing. The Nanodrop [78], Qubit 2.0 [79], and Aglient 2100 [80] methods were used to verify the purity, concentration, and integrity of the RNA samples to ensure that sufficiently high-quality samples were used for transcriptome sequencing.

After testing and qualifying the samples, the library was constructed according to the following procedures. First, eukaryotic mRNA was enriched with oligo (dT)-coated magnetic particles [81]. Fragmentation buffer was then added to interrupt the mRNA in a random manner. The first cDNA strand was then synthesized using mRNA as a template and six-base random hexamers, followed by the addition of buffer, dNTPs, RNase H, and DNA polymerase I to synthesize the second cDNA strand [82]. cDNA was purified with the aid of AMPure XP particles. Then, end repair, A-tail addition, and sequencing junction ligation were performed. The fragment size was then selected using AMPure XP beads [83]. Finally, the cDNA library was enriched via PCR.

### 4.2. Library Quality Management, Data Sequencing and Quality Assurance

After library construction, the library concentration and insert size were analyzed using Qubit 2.0 and Agilent 2100, respectively, and the effective library concentration was quantified using the Q-PCR method [84]. After ensuring qualification, Illumina Hiseq high-throughput sequencing was performed [82].

Based on sequencing-by-synthesis (SBS) technology, the cDNA libraries were sequenced using the Illumina HiSeq high-throughput sequencing platform, which can generate a large number of high-quality reads [85]. These reads or bases generated by the sequencing platform are referred to as raw data, and the majority of their base quality scores can meet or exceed Q30 (99.9% identification accuracy).

Before continuing with the subsequent analysis, it was necessary to ensure that the reads used were of sufficient quality to ensure the accuracy of sequence assembly and the subsequent analysis. In addition, there were very few reads containing sequencing primers, connectors, and other artificial sequences in the raw data, so it was necessary to truncate the sequencing connectors and primer sequences in the reads and filter the low-quality data to ensure data quality [86].

### 4.3. Transcriptome Sequencing Data Compilation

After obtaining high-quality sequencing data, the sequence was assembled. Trinity [87] is a transcriptome sequencing assembly software designed for high-throughput sequencing. The sequencing depth of a transcript is influenced by the quantity of sequencing data and other factors, as well as the abundance of the transcript’s expression. The sequencing depth has a direct impact on the quality of the assembly [88].

In the beginning, the k-mer library was created by breaking the sequencing reads into shorter k-mers using the Trinity software and removing any k-mers that contained errors. Next, the k-mer with the highest frequency was chosen as the seed for greedy extension at both ends, and the process was repeated until the k-mer library was exhausted [89]. Next, the contigs were clustered to produce components, the De Bruijn graph was constructed for each component, and the nodes and edges were merged and trimmed to simplify the De Bruijn graph [90]. Finally, the De Bruijn graph [91] was deconstructed using the actual read in order to identify the transcribed sequences in each fragment collection independently.

### 4.4. Plant Materials in Populations

The ETS-SSR markers in plants were used to assess the efficacy of ex situ *A. nanus* conservation efforts. Plant samples were collected from Wuqia (50) and an ex situ reserve (24), as well as Kyrgyzstan (13), to identify genetic diversity in different regions and protect germplasm resources (Table 11).

### 4.5. DNA Extraction

Sampled plant materials were stored in liquid nitrogen following field collection. After transport to the laboratory, about 0.25 g fresh leaves was ground into a powder in liquid nitrogen. Next, the CTAB method was used to extract their DNA following standard procedure. The concentration was measured by spectrophotometry, diluted to about 0.10–0.25 mg/µL, and refrigerated at −20 °C for later use [90].

### 4.6. PCR Amplification and EST-SSR Primers Selection

The primers and DNA template described above were used for PCR reaction. The PCR reaction system was 25 µL, as follows: ddH_2_O 9.5 µL, Mix12.5 µL, primers at both ends 1 µL, and DNA1 µL. The PCR reaction process was as follows: denaturing at 94 °C for 2 min; 30 cycles at 94 °C for 1 min, annealing temperature 53–56 °C for 1 min and 72 °C for 2 min; and 72 °C for 5 min. The PCR products were subjected to 1% agarose gel electrophoresis using an electrophoresis system running the buffer solution of 1 × Tris EDTA (TE), with a constant voltage of 130 V and an electrophoresis time of 15 min. Successful amplifications (with the clear band) were sent to Shanghai SBS, Biotech Ltd. for analysis. Then, 15 pairs (Table 12) of 2–5 base SSR sequences with 3–4 repeats (about 22 bases) were selected for primer synthesis by a bio-company (Sangon, Shanghai, China) with 5′ fringe HEX fluorescence labeling.

### 4.7. Statistical Analyses

MISA [92] is a Perl-based script that identifies SSRs from FASTA sequences and has become the tool of choice for most EST-SSR researchers due to its ease of use, lack of networking, minimal hardware requirements, and compatibility with multiple tools.

GeneMarker [93] was used to analyze the original data obtained using high-performance capillary electrophoresis (HPCE). GenAlEx [94] was used to calculate the population genetic parameters of each primer pair, including the number of alleles per locus (Na), observed heterozygosity (Ho), expected heterozygosity (He), effective number of alleles (Ne), and information index and percentage of polymorphic sites (PPB) values. Principal component analysis (PCoA) was also performed in GenAlEx. Additionally, an analysis of molecular variance (AMOVA) function was adhibited in GenAlEx to measure ex situ conservational efficacy for *A. nanus* within and among populations, and within and among groups.

In total, 999 random permutations were implemented in GenAlEx for each sample. The genetic structures of three (Wuqia, Kyrgyzstan, and ex situ reserve) and two (Wuqia, Kyrgyzstan, no ex situ reserve) natural populations of *A. nanus* were analyzed based on Bayesian analysis using the STRUCTURE software (version 2.3.4) [95]. Ten independent runs were performed for each K value. Then, results were submitted to the online software Structure Harvester to analyze the optimal K value, which was determined by the method described by the burn-in period iterations and Markov chain Monte Carlo repetitions. Finally, Core Finder was used to calculate the size of individual protected germplasm resources.

## Figures and Tables

**Figure 1 plants-12-02670-f001:**
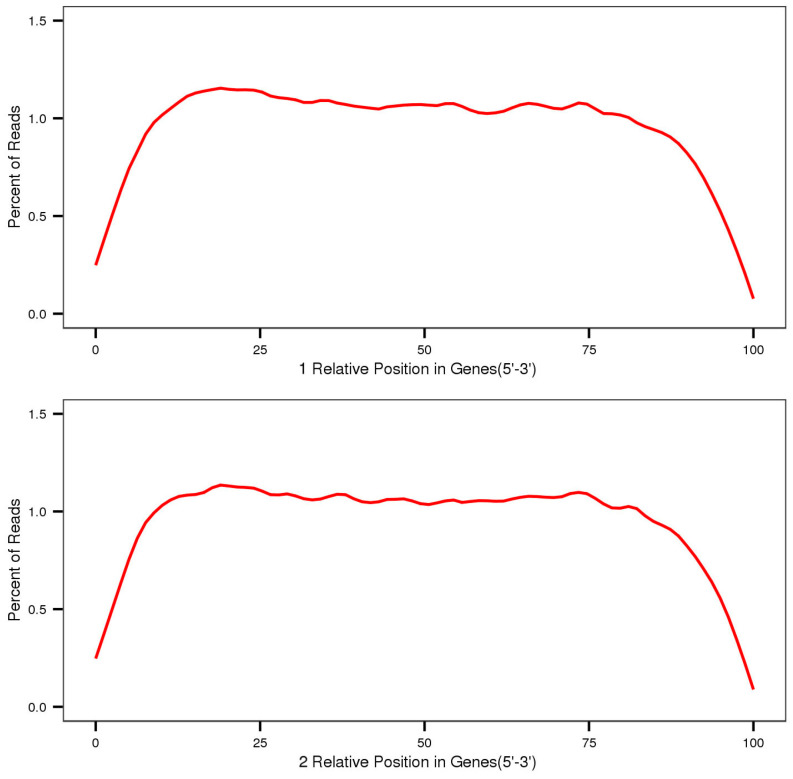
Positional distribution of mRNA mapped reads of *A. nanus*.

**Figure 2 plants-12-02670-f002:**
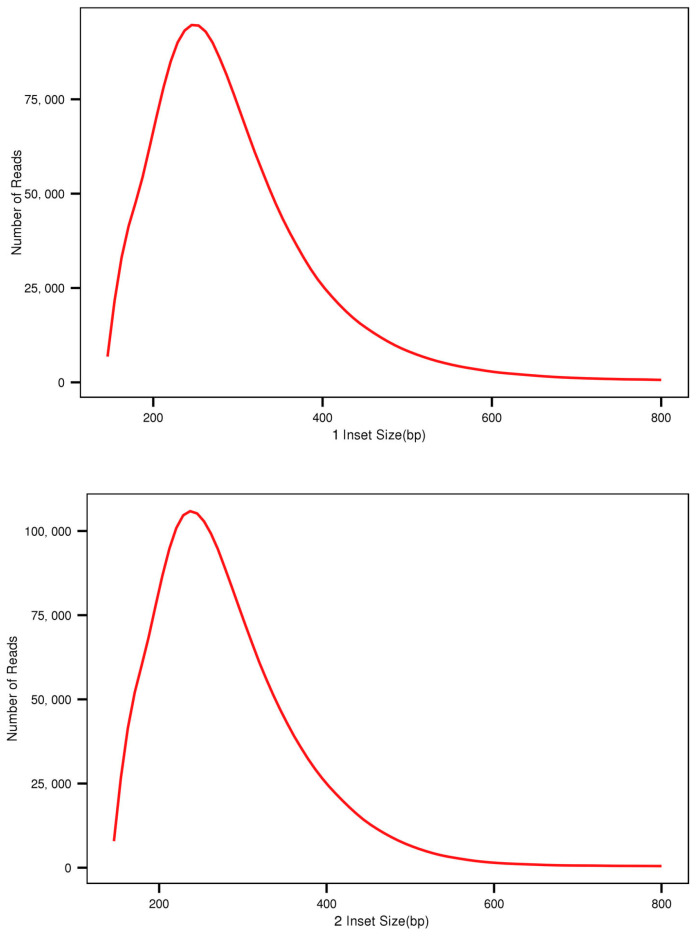
Simulation of insert fragment length distribution of *A. nanus*.

**Figure 3 plants-12-02670-f003:**
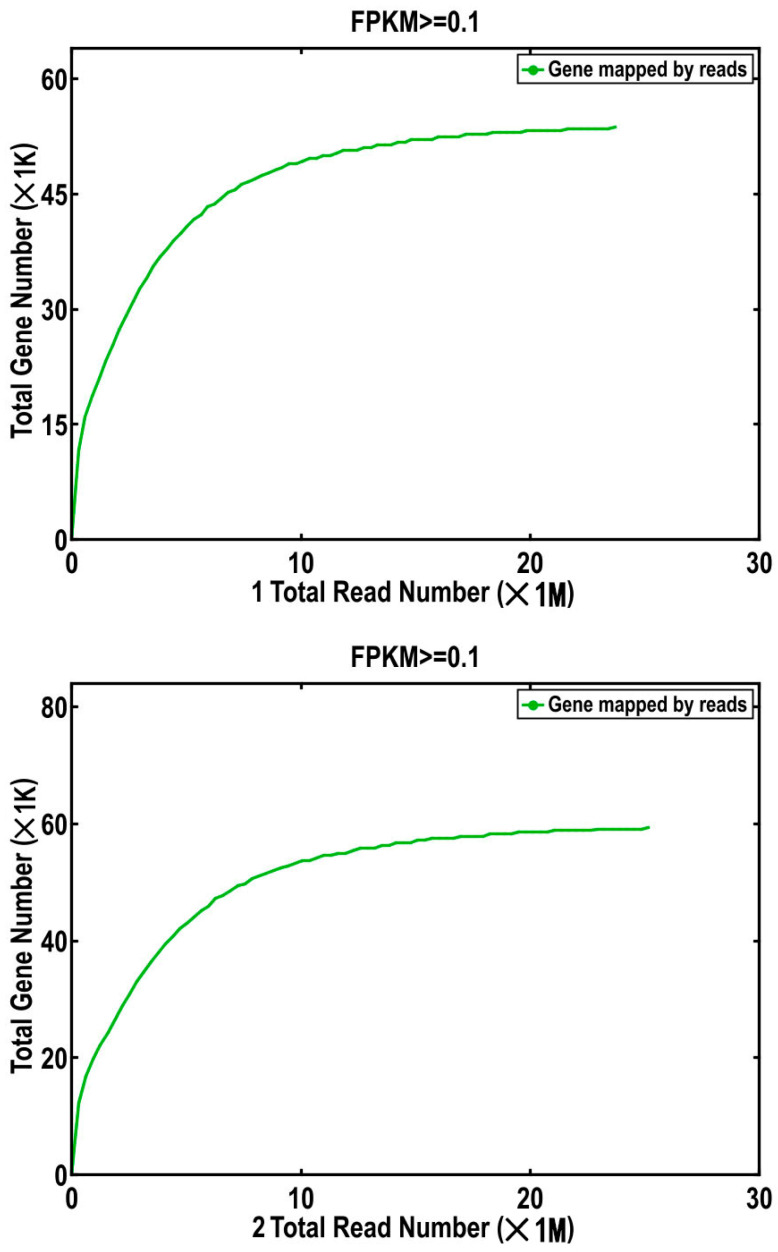
Simulation of transcriptome sequencing data saturation of *A. nanus*.

**Figure 4 plants-12-02670-f004:**
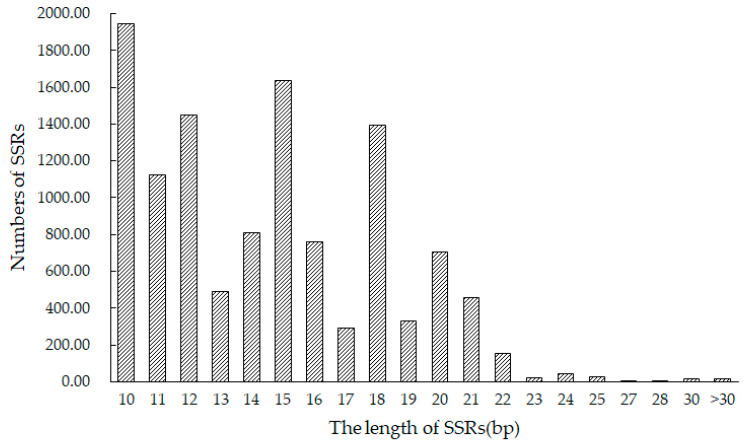
Length distribution of SSRs in the *A. nanus* transcriptome genome.

**Figure 5 plants-12-02670-f005:**
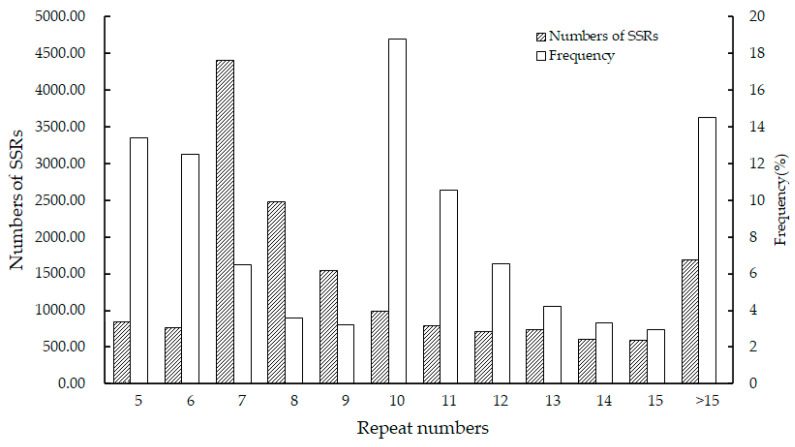
Distribution of repeat numbers of SSRs in the *A. nanus* transcriptome genome.

**Figure 6 plants-12-02670-f006:**
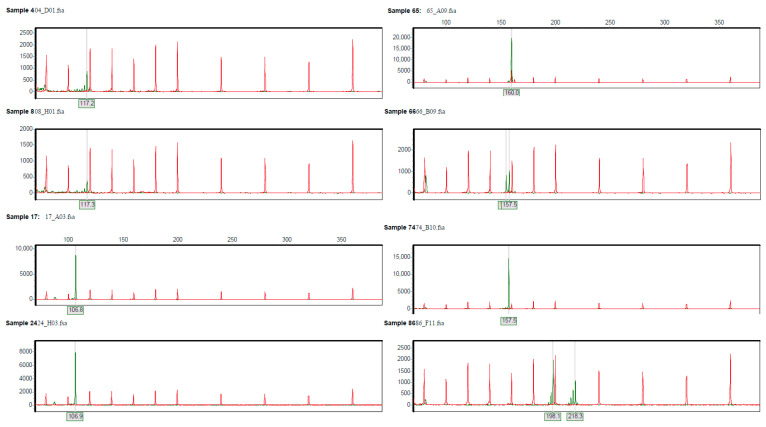
Partial genotyping peak plots of polymorphic SSR primers for the representative samples of *A. nanus*.

**Figure 7 plants-12-02670-f007:**
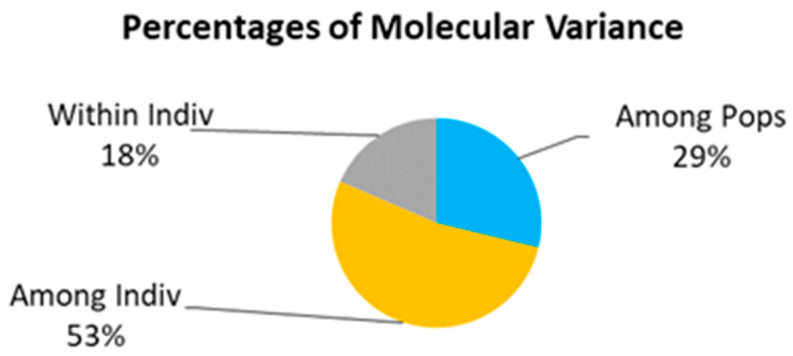
The percentage of molecular variance in genetic variation within and between *A. nanus* populations.

**Figure 8 plants-12-02670-f008:**
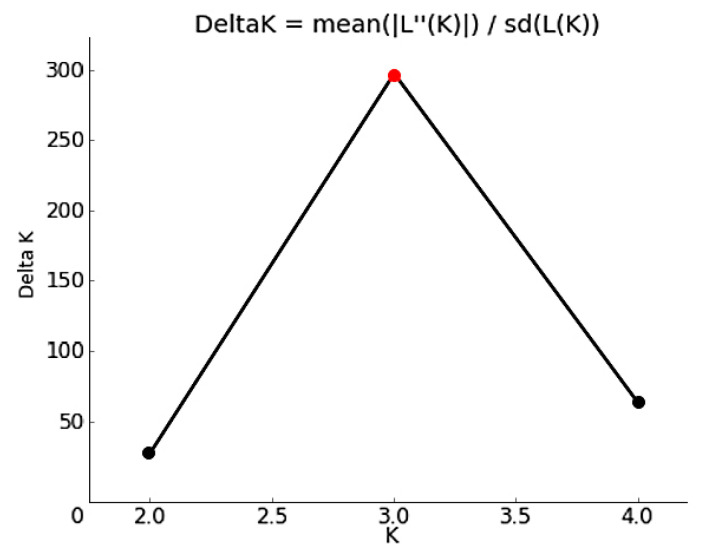
Genetic structure analysis of *A. nanus* based on Bayesian model.

**Figure 9 plants-12-02670-f009:**
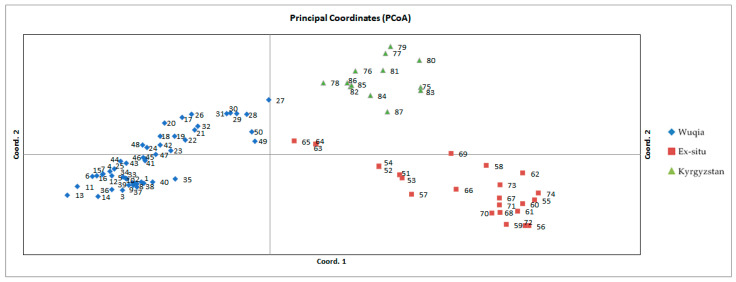
Principal coordinate analysis of three groups of *A. nanus* based on SSR genetic distance. The different colors and shapes represent different study populations: blue diamonds for the Wuqia population, red squares for the ex situ population, and green triangles for the Kyrgyzstan population. The axis labels refer to the percentage of explained variance for each principal coordinate. The different numbers for each sample are listed in Table 11.

**Table 1 plants-12-02670-t001:** Sample sequencing data evaluation statistics of *A. nanus*.

Samples	Read Number	Base Number	GC Content	% ≥ Q30
*A. nanus* ^1^	23,712,117	7,072,497,166	44.94%	92.77%
*A. nanus* ^2^	25,148,183	7,496,125,488	45.26%	92.86%

^1,2^ Samples: sample name; Read Number: total number of pair-end Reads in clean data; Base Number: total number of bases in clean data; GC Content: clean data GC content, which refers to the ratio of G and C bases to the total number of bases in the clean data; % ≥ Q30: percentage of bases with clean data quality value greater than or equal to 30.

**Table 2 plants-12-02670-t002:** Comparison table of sequencing data and assembly results of *A. nanus*.

Samples	Clean Reads	Mapped Reads	Mapped Ratio
*A. nanus* ^1^	23,712,117	18,438,968	77.76%
*A. nanus* ^2^	25,148,183	19,535,876	77.68%

^1,2^ Samples: sample name; Clean Reads: number of Clean Reads, counted on both ends; Mapped Reads: number of Mapped Reads, counted on both ends; Mapped Ratio: ratio of Mapped Reads to Clean Reads.

**Table 3 plants-12-02670-t003:** Comparison table of sequencing data and assembly results.

Anno Database	Annotated Number	300 ≤ Length < 1000	Length ≥ 1000
Nr Annotation	37,530	13,665	16,236
Swissprot Annotation	21,475	7073	11,007
GO Annotation	23,157	7830	10,705
COG Annotation	9093	2285	5664
KOG Annotation	21,174	7081	9841
eggnog Annotation	34,853	12,261	15,588
KEGG Annotation	11,687	3901	5707
Pfam Annotation	21,979	6245	12,883
Nr Annotation	37,530	13,665	16,236

**Table 4 plants-12-02670-t004:** Overview of SSRs in the *A. nanus* transcriptome genome.

Items	Numbers
Total size of transcriptome genome (Mb)	42.52
Total number of identified SSRs	11,645
Total length of SSRs (bp)	169,920
Frequency (SSRs/Mb)	273.88
Density (bp/Mb)	3996.48
Total content of transcriptome genome SSRs (%)	0.40

**Table 5 plants-12-02670-t005:** SSR markers analysis of *A. nanus*.

Repeat Type	Predominant Type	Number	Proportion (%)	Frequency (SSRs/Mb)	Total Length (bp)	Average Length (bp)
Mono	A/T	6731	57.80	158.31	88,629	13.17
Di	AG/TC	2231	19.16	52.47	34,494	15.46
Tri	GAA/AAT	2375	20.40	55.86	39,996	16.84
Tetra	AAAT/ATTT	250	2.15	5.88	5088	20.35
Penta	-	33	0.28	0.78	855	25.91
Hexa	-	25	0.21	0.59	858	34.32
Total		11,645	100	273.89	169,920	14.59

**Table 6 plants-12-02670-t006:** Main motif of *A. nanus* transcriptome genome SSRs.

The Motif of Repeat	Repeat Numbers						Total	Percentage (%)
	5	6	7	8	9	>9		
A/T	0	0	0	0	0	6505	6505	55.86
AG/TC	0	236	131	142	117	109	735	6.31
AT/CT	0	163	121	102	103	97	586	5.03
TA/GA	0	168	83	113	103	91	558	4.79
AAT/GAA	127	71	37	2	0	0	237	2.04
AGA/CTT	90	66	37	0	0	0	193	1.66
ATT/TTC	97	56	29	1	0	0	183	1.57
AAG/TCT	78	70	19	0	0	0	167	1.43
ATA/TTA	73	46	23	3	0	0	145	1.25
TAT/GTT	68	44	23	0	0	0	135	1.16
TTG/CAA	72	31	13	0	1	0	117	1.00
ACA/TCA	51	38	14	0	0	0	103	0.88
AAAT/ATTT	18	0	0	0	0	0	18	0.15
AGAA/AGAT	13	1	0	0	0	0	14	0.12
TATT/TCAT	11	1	0	0	0	0	12	0.10
AAGG/ATAA	9	2	0	0	0	0	11	0.09
Total	707	993	530	363	324	6,802	9,719	83.46
Percentage (%)	6.07	8.53	4.55	3.12	2.78	58.41	83.46	

**Table 7 plants-12-02670-t007:** Genetic diversity of the 15 SSRs primers.

Locus	Ht	Mean He	Mean Ho	Fis	Fit	Fst	Nm
F1	0.899	0.698	0.282	0.595	0.686	0.224	0.868
F2	0.854	0.228	0.000	1.000	1.000	0.733	0.091
F3	0.902	0.713	0.265	0.628	0.706	0.210	0.942
F4	0.816	0.447	0.167	0.627	0.796	0.452	0.303
F5	0.815	0.581	0.550	0.055	0.326	0.287	0.622
F6	0.904	0.716	0.379	0.471	0.581	0.208	0.951
F7	0.772	0.315	0.292	0.073	0.621	0.592	0.173
F8	0.806	0.419	0.074	0.823	0.908	0.481	0.270
F9	0.721	0.218	0.222	−0.021	0.692	0.698	0.108
F10	0.691	0.292	0.072	0.752	0.895	0.578	0.182
F11	0.861	0.250	0.218	0.129	0.747	0.710	0.102
F12	0.818	0.481	0.098	0.797	0.881	0.412	0.357
F13	0.832	0.497	0.476	0.042	0.428	0.403	0.370
F14	0.821	0.462	0.203	0.562	0.753	0.437	0.323
F15	0.889	0.670	0.515	0.231	0.420	0.246	0.764
Mean	0.83	0.47	0.25	0.45	0.70	0.44	0.43

**Table 8 plants-12-02670-t008:** Genetic diversity parameters of *A. nanus* populations.

Pop	N	Na	Ne	I	Ho	He	uHe	F	PPB
Wuqia	31.867	6.333	2.598	1.121	0.29	0.535	0.552	0.514	0.9333
Ex situ	16.667	5	2.714	0.957	0.231	0.453	0.476	0.437	0.8
Kyrgyzstan	7.533	3.6	2.429	0.827	0.242	0.409	0.449	0.456	0.7333
Mean	18.689	4.978	2.58	0.969	0.254	0.466	0.492	0.472	0.8222
Total	56.067	14.933	7.741	2.905	0.763	1.397	1.477	1.407	2.4666

**Table 9 plants-12-02670-t009:** Genetic distances and genetic differentiation coefficients of *A. nanus* populations.

Nei Genetic Distance	Wuqia	Ex Situ	Kyrgyzstan	FST Values	Wuqia	Ex Situ	Kyrgyzstan
Wuqia	0.000			Wuqia	0.000		
Ex situ	4.299	0.000		Ex situ	0.352	0.000	
Kyrgyzstan	4.909	2.659	0.000	Kyrgyzstan	0.405	0.431	0.000

**Table 10 plants-12-02670-t010:** Analysis of molecular variance (ANOVA) of genetic variation within and among groups of *A. nanus*.

Source	DF	SS	MS	Est. Var.	Proportion
Among Pops	2	211.259	105.629	1.956	29%
Among Indiv	84	707.678	8.425	3.583	53%
Within Indiv	87	109.5	1.259	1.259	18%
Total	173	1028.437		6.797	1

**Table 11 plants-12-02670-t011:** Population information of *A. nanus*.

Pop	Number	Latitude	Longitude	Elevation
Wuqia	1~50	39.7° N	75.01° E	2181 m
Ex situ	51~74	39.72° N	75.27° N	2178 m
Kyrgyzstan	75~87	41.71° N	74.34° E	1698 m

**Table 12 plants-12-02670-t012:** EST-SSR primers sequence information of *A. nanus*.

SSR Primer Pair	SSR Primer Sequence (5′-3′)	Fluorescence Tag
c11315	F:ACTGCACCTGTCCAGATTCC	5′HEX
	R:TTGATTCGCTGTCTCCCTCT	
c23139	F:AGCATGCATGTGCCTTTTTA	5′HEX
	R:ATCGGGAGAGCAACAGTACG	
c23483	F:AACGCGTCCCATTCTCTCT	5′HEX
	R:TCATCTTTCAAAAGGGGCAC	
c23540	F:GATCAGCTTCTCACCCGAAG	5′HEX
	R:TCTGGGTCTCTTGGCCTCTA	
c24361	F:AATGCATGATTGATTGTCACTG	5′HEX
	R:TTGCGGACTGCAACAATTAG	
c24982	F:CCGTCTTGCAGGTTTCAAGT	5′HEX
	R:TTACCGGCCATAGATTGAGG	
c32449	F:TCAATCTTCGTAGCTTTCCCA	5′HEX
	R:GCTCCAAAGTCCACCACATT	
c34776	F:CCACTTTGGCTCATTGCTTT	5′HEX
	R:CTGAAGGGAGTGTGGCATTT	
c34960	F:GACCCATCAAAACATGACCC	5′HEX
	R:GGACCTCAGGATGGTCAAGA	
c35322	F:TTGTTACAAAAGCCGATCCC	5′HEX
	R:TAGGAATGACTCCCACTGCC	
c44083	F:TGAAGAGACGCAATGGTGAG	5′HEX
	R:ATGCCCAACTTGGATAGTGC	
c44298	F:AGCTGTCTTCCCAGTACCGA	5′HEX
	R:ATGGTGAGCACAACCATGAA	
c44674	F:TAATTGACCTCGAACCTCCG	5′HEX
	R:TGGAGGAGAAGGGGGTTACT	
c52855	F:GAAAGGTTGGGGATTTTGGT	5′HEX
	R:AGTGGGTGAGGGTGAAACTG	
c52997	F:CAATATCCAAAGGTGGGGTG	5′HEX
	R:GCTACGTCTCGTGCTAGCCT	

## Data Availability

The data presented in this study are available on request from the corresponding author.

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
