# Peer review of "Identification of the Efficacy of Ex Situ Conservation of Ammopiptanthus nanus Based on Its ETS-SSR Markers"

_plants, 2023, doi:10.3390/plants12142670_

Round 1

Reviewer 1 Report

1.       Provide detailed results such as assembly, GO annotation, etc. for both transcriptome analyses.

2. Clarify whether both transcriptomes were merged for EST-SSR mining.

3. Since EST-SSRs are less polymorphic, therefore 15 makers are not sufficient for a diversity study and suggesting a conservation strategy.

The English language requires attention because there are grammatical errors and typographical mistakes.

Author Response

Dear Reviewer 1:

We greatly appreciate your useful comments and have responded to each of them as follows:

  1. Provide detailed results such as assembly, GO annotation, etc. for both transcriptome analyses.

An: We appreciate this constructive suggestion and have made the changes in the manuscript. The Transcriptome sequencing library quality assessment and Annotation of Unigene function have been added to the Results to supplement the previous information. See them in Line 227-317, p6-p10.

  1. Clarify whether both transcriptomes were merged for EST-SSR mining.

An: We have revised the description of the original transcriptomes’ data in the Methods section in order to clarify that just one A. nanus dataset has been used in the MS. Please see the edits in Line119-120, p3.

  1. Since EST-SSRs are less polymorphic, therefore 15 makers are not sufficient for a diversity study and suggesting a conservation strategy.

An: We agree with you that EST-SSRs are less polymorphic than them in the whole genome data, so we added an analysis of the polymorphism of ten communities from three populations. To the extent that the results represent a portion of broader biodiversity patterns (Figure 1 in the Review Report), and in conjunction with our planting work in botanical gardens, we also can make effective conservation recommendations for the current issues identified. To further improve our conservation project, we would like to introduce additional markers and data in analyses of A. nanus conservation in future studies.

Lastly, we apologize for this grammatical error and thank you very much for identifying it. We have made the necessary corrections based on your feedback.

Sincerely yours,

Wang Jian-Cheng & Shi Wei

E-mail: www-1256@ms.xjb.ac.cn (WJC) shiwei@ms.xjb.ac.cn (SW)

Tel: +86-991-7885307; Fax: +86-991-7885320

Postal address: Xinjiang Institute of Ecology and Geography, Chinese Academy of Sciences, 818 South Beijing Road, Urumqi, Xinjiang, China; Post code: 830011

Reviewer 2 Report

The paper is interesting, very technically relevant for conservation and promoting ecosystem services in an arid environment. However, it would be useful to integrate this study into the broader context of potential for reintroduction into its habitat. Therefore, in the Discussion, it would be important to recommend undertaking studies on GxE interactions and the linking morphological and molecular diversity, providing examples where this has been already undertaken successfully. 

There are significant errors in English language and scientific/technical writing conventions - these should be addressed prior to resubmitting. 

Author Response

Dear Reviewer:

We greatly appreciate your key suggestion regarding the rare and endangered plants conservation, and the corresponding content has been added in the manuscript.

Responses to the reviewers' comments:

The paper is interesting, very technically relevant for conservation and promoting ecosystem services in an arid environment. However, it would be useful to integrate this study into the broader context of potential for reintroduction into its habitat. Therefore, in the Discussion, it would be important to recommend undertaking studies on GxE interactions and the linking morphological and molecular diversity, providing examples where this has been already undertaken successfully.

An: We agree that discussion of the GxE interactions and molecular diversity should be strengthened in this manuscript, especially in populations of A. nanus in Kyrgyzstan and China. Thus, in both the introduction and discussion sections, we have added content about the relationship between genes and the environment, as well as relevant supporting references.

Sincerely yours,

Wang Jian-Cheng & Shi Wei

E-mail: www-1256@ms.xjb.ac.cn (WJC) shiwei@ms.xjb.ac.cn (SW)

Tel: +86-991-7885307; Fax: +86-991-7885320

Postal address: Xinjiang Institute of Ecology and Geography, Chinese Academy of Sciences, 818 South Beijing Road, Urumqi, Xinjiang, China; Post code: 830011

Round 2

Reviewer 1 Report

Accept.

English language needs attention.